# Learning Causal Representations with Granger PCA

**Gherardo Varando**[1]     **Miguel-Ángel Fernández-Torres**[1]     **Jordi Muñoz-Marí**[1]     **Gustau Camps-Valls**[1]

[1]Image Processing Laboratory (IPL), Universitat de València, València, Spain

## Abstract

Learning causal feature representations helps us identify relevant subspaces to express the signal of interest and understand (and imagine interventions on) the underlying causal mechanisms. In this work, we adopt a rather pragmatic standpoint and propose learning Granger-causal feature representations with a simple additional rotation on top of the classical Principal Component Analysis (PCA). We generalize the methodology to nonlinear Granger causal representations with kernel PCA, give empirical proof of performance in linear and nonlinear toy examples, and find the relevant problem of finding Granger-causal feature long-range spatio-temporal teleconnections in the Earth system. The methodology can be of practical convenience in high-dimensional and low-sample sized problems.

## 1  INTRODUCTION

Causal discovery and inference using high-dimensional time processes, e.g. spatio-temporal observations of gridded variables, typically requires a first step of dimensionality reduction. This is the case when working with fMRI recordings [Zhou et al., 2009, Wen et al., 2013] or with Earth observation data and climate model outputs [Runge et al., 2015, Nowack et al., 2020]. Principal Components Analysis (PCA) is undoubtedly the preferred method in real practice but, since the components are not necessarily physically meaningful, PCA is followed by a varimax (or oblique) rotation [Lian and Chen, 2012, Hendrickson and White, 1964, Kaiser, 1958] to extract relevant (spatio-temporal) components [Vejmelka et al., 2015]. Next, a causal discovery algorithm, e.g. Granger causality [Granger, 1969], the classical PC algorithm [Spirtes et al., 2000], or PC variants like PCMCI [Runge et al., 2019] are applied to the extracted

time series to estimate a causal graph. Unfortunately, the feature extraction is generally disconnected and learned independently from the causal task, which leads to a quite arbitrary selection of the modes of variability explaining the phenomena the data is representing.

Recently, Causal Representation Learning [Schölkopf et al., 2021] has been framed as a fundamental problem in the intersection of machine learning and causal inference. However, the task of identifying suitable high-level-representations of fine-grained observations that admit causal models is challenging for both human and machine intelligence. Yet, it is fundamental within the general goal of modern machine learning to learn meaningful representations of data. Actually, the main stress so far has been to learn disentangled, non-spurious and efficient representations of the data [Wang and Jordan, 2021]. In particular, the overarching goal in the representation learning framework is about learning meaningful *causes* in the data generation causal model.

In this work, we instead focus on learning high-level representations that are *effects* of known low-dimensional causes. This is an important endeavour in domains where high-dimensional (e.g. spatio-temporal) observations are used, and where the goal is not only to reconstruct well the signal but recovering lower-dimensional representations that correspond to causal effects of known drivers. We introduce Granger PCA (GPCA), a methodology to learn Granger-causal feature representations (§2). The method is simple, only implies an additional rotation on top of the PCA one, and can be generalized to nonlinear cases (e.g. using kernel PCA [Scholkopf et al., 1998]). Relations and advantages over other multivariate analysis methods like PLS or CCA [Arenas-García et al., 2013] are outlined in §3. We give empirical proof of performance in linear and nonlinear toy examples (§4.1), and also in the relevant problem of finding Granger-causal long-range spatio-temporal connections in the Earth system (§4.2). We anticipate a wide use and adoption of GPCA in scientific domains where dimensionality reduction and Granger causality are combined.

*Accepted for the Causal Representation Learning workshop at the 38th Conference on Uncertainty in Artificial Intelligence* (UAI CRL 2022).

## 2 GRANGER PCA

The methodology proposed to extract principal components that are causal is described here. We start explaining how to find an optimal rotation that aligns axes with causally meaningful directions when a sufficient number of observations is available. We then describe how to generalize this to realistic high-dimensional cases thanks to a first whitening step with classical PCA and easily generalize the methodology to nonlinear cases.

### 2.1 PROBLEM SETTING AND NOTATION

Consider two time-dependent processes $X(t)$ and $Y(t)$, where $X(t) \in \mathbb{R}^d$ and $Y(t) \in \mathbb{R}^p$ for each $t$, where we assume $d \gg p$. We will mainly consider the case $p = 1$, but the proposed approach can be extended to $p > 1$. For the sake of notation simplicity we assume uniform time sampling and thus denote that we observe processes $X(t)$ and $Y(t)$ at $n$ times $t_i = i\Delta t$. The observation (design) matrices are denoted as $X \in \mathbb{R}^{n \times d}$ and $Y \in \mathbb{R}^{n \times p}$, with entries $X_{ij} = X_j(t_i)$ (likewise for $Y$). The goal is to find *components* of $X(t)$ which are causally related with $Y(t)$ and, in particular, we will consider linear Granger causality [Granger, 1969] in what follows.

### 2.2 DIRECTIONS OF MAXIMAL CAUSALITY

We recall that a time-series $A(t)$ is linearly Granger-cause of $B(t)$ with respect to a fixed time-lag $m$ if the null hypothesis $\{\alpha_\ell = 0 \text{ for } \ell = 1, \ldots, m\}$ is rejected for the linear auto-regressive (AR) model $B(t) = \sum_{\ell=1}^{m} \beta_\ell B(t - \ell) + \sum_{\ell=1}^{m} \alpha_\ell A(t - \ell) + \beta_0 + \epsilon$. That is $A$'s past jointly adds explanatory power to the auto-regressive model for $B$. The above null hypothesis is usually tested by comparing the residual sum of squares (RSS) of the full and the restricted ($\alpha_i = 0$ for all $i$) models, for instance using an $F$-test.

We simply define a Granger rotation as the orthogonal linear transformations of $X$ that transforms the data into a rotated coordinate system such that the first coordinate comes to be the one for which $Y$ is more Granger causal, and so on for the next components. Let $U$ be the orthogonal matrix representing such transformation and let $u_{(i)}$ denote its $i$-th column and $\tilde{X} = XU$ be the transformed data. Now the goal is to reduce the RSS of the auto-regressive models including the past of $Y(t)$ ($RSS_1$), and also to increase the RSS of the restricted auto-regressive models by considering just $\tilde{X}(t)$ ($RSS_0$). In particular, we search the orthogonal transformation that maximizes the difference between the two residual sum of squares, $RSS_0 - RSS_1$. For the case of extracting only one feature, this problem can be solved by maximizing the following Rayleigh quotient

$$u_1 = \arg\max_u \left\{ \frac{(u^\top X^\top Q_2 Q_2^\top X u)}{u^\top u} \right\},$$

where $Q = [Q_1 | Q_2 | Q_3]$ is the $Q$ term in the QR-factorization of the matrix $[X_{past} | Y_{past}]$ obtained by stacking the lagged observations of $X$ and $Y$ (up to a certain maximum lag $m$).

As in PCA more components can be extracted by Hotelling's deflation. One can subtract the first $k$ components and repeat the process above to obtain the $k+1$ column of $U$. The resulting $U$ is composed of columns which are the eigenvectors of $(X^0)^T (W^T W - V^T V) X^0$ sorted by the corresponding eigenvalues. The full matrix of the orthogonal transformation $U$ consists of the eigenvectors of $X^\top Q_2 Q_2^\top X$ or, equivalently, the singular vectors of $Q_2^\top X$. We refer the reader to Appendix A for details and derivations.

### 2.3 ROTATED PRINCIPAL COMPONENTS

The introduced Granger rotation $U$ is well defined for observations $X \in \mathbb{R}^{n \times d}$, $Y \in \mathbb{R}^{n \times d}$ and a given maximum lag $m > 0$, if $n - m > dm$. For the case of high-dimensional data, where $n - m < dm$, we propose to combine the defined Granger rotation after whitening and reducing the data with PCA or, alternatively, a non-linear extension such as kernel PCA. The method consists in replacing $X$ with the projection onto the first $k$ principal components and then applying the Granger rotation as defined previously. We refer to such procedure as Granger-rotated PCA (GPCA) or Granger-rotated kernel PCA (G-kPCA), if we make use of the nonlinear version based on kernel PCA as first step [Scholkopf et al., 1998].

## 3 RELATED METHODS

The proposed approach is tightly related to various classical (un)supervised dimensionality reduction techniques, especially in the field of multivariate data analysis [Arenas-García et al., 2013], which we briefly review here.

PCA [Pearson, 1901, Hotelling, 1933, Jolliffe, 2003], eventually followed by an additional rotation, has been used extensively for extracting principal modes of variability in high-dimensional data. In geophysics, where spatio-temporal data needs to be compressed and visualized, PCA is traditionally referred to as Empirical Orthogonal Functions (EOFs) and its different variants proposed [Preisendorfer, 1988, Bauer-Marschallinger et al., 2013, Volkov, 2014, Forootan et al., 2016]. As we already discussed in Section 1, PCA/EOF is supplemented with an extra varimax rotation [Kaiser, 1958] that seeks more meaningful principal 'modes of variability' (i.e. spatial regions of maximum variance) previous to causal discovery [Vejmelka et al., 2015].

Instead of adding extra rotations, and for supervised settings, one modifies the problem by replacing variance with correlation or covariance maximization, such as in Canonical Correlation Analysis (CCA) [Hotelling, 1936] and Partial

Least Squares (PLS) [Wold, 1966], respectively. These are indeed similar methods to the proposed approach, as they search for a latent space that represents the data well and is related to an external signal. In particular, CCA has been extended to the temporal domain [Bießmann et al., 2010] by considering time-lagged copies of the variables. Similarly, PLS can be applied to time-series observations by considering lagged copies of each component and thus searching for maximally correlated latent spaces. We will adapt these methods for comparison in our experiments.

## 4 EXPERIMENTAL RESULTS

We illustrate the performance of the GPCA in a simulation study with generated data from a known dynamical structural equation model (both in low and high dimensional settings) and in a challenging Earth system science problem. The code to reproduce the experiments is available at https://anonymous.4open.science/r/GrangerPCA-3B7E/.

### 4.1 SIMULATION EXPERIMENTS

We simulate observations from a linear dynamical structural equation model, whose summary graph is depicted in Fig. 1. Moreover, a detailed specification is given in the Appendix B.2. The system is composed by a multi-dimensional process $X(t) \in \mathbb{R}^d$, in particular $X(t) = (X_A, X_B, X_C, X_D, X_E)(t)$, where $|A| = \ldots = |E| = d/5$. As depicted in Fig. 1, only variables $X_A$ are direct effects of the exogenous process $Y$, while autocorrelation, indirect effects on downstream variables and non-observed confounding processes are present.

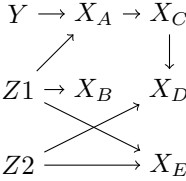

Figure 1: Summary graph of the dynamical structural equation model for the simulated data. All $X$ variables are also dependent on their own past values.

We sample $n = 1000$ observations at times $t = 1, \ldots, 1000$ and thus $X \in \mathbb{R}^{1000 \times 10}$, $Y \in \mathbb{R}^{1000}$. Then, we compare the components extracted by classical PCA, Granger-rotated PCA (GPCA), time-lagged CCA (TLCCA) and time-lagged PLS (TLPLS). We consider $X$ centered and scaled to unit variance and use a maximum lag of 5 for all methods. We then compare the approaches by visualizing the obtained rotation matrices in Fig. 2 and by computing the Granger causality test of each learned component with the causal signal $Y(t)$ (see $p$-values in Table 1). The automatic causal

ordering of GPCA is quite convenient in practice as it leads to richer feature extractions, cf. Fig. 3.

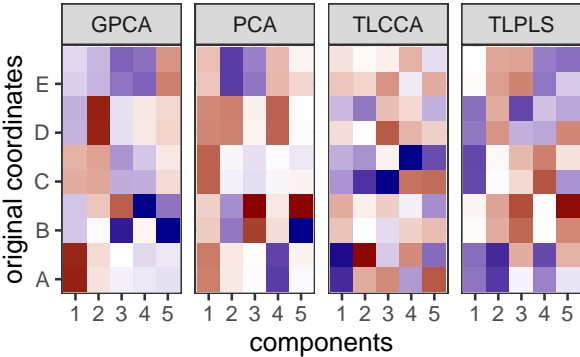

Figure 2: Rotations learned with various methods ($d = 10$).

Table 1: Granger causality test $p$-values for the first five components for the different methods ($d = 10$).

|    | PC | GPC | TLCCA | TLPLS |
|----|------|-----------|----------|----------|
| C1 | 6.50e-19 | 1.79e-103 | 6.82e-90 | 3.29e-13 |
| C2 | 2.06e-01 | 7.22e-02 | 6.96e-72 | 1.10e-69 |
| C3 | 3.04e-02 | 3.46e-01 | 5.75e-08 | 1.88e-01 |
| C4 | 5.10e-73 | 6.38e-01 | 3.86e-03 | 3.11e-51 |
| C5 | 2.08e-01 | 9.82e-01 | 1.99e-01 | 2.95e-01 |

**Optimality of GPCA extracted components** The learned first components and the average signal of the $X_A$ components $X_m(t) = 1/|A| \sum_{i \in A} X_i(t)$ are plotted in Fig. 3. The mean signal $X_m(t)$ is by construction a Granger-effect of $Y(t)$ with maximum time-lag equal to 3. We can see that the *causal* first component learned with GPCA correctly identifies a linear combination of $X_A(t)$ as the most causally related with the target signal $Y(t)$. PCA is unable to isolate the causal signal, and we can observe how the principal components are not useful for causal discovery in this example. TLCCA correctly recovers the first component which is somehow similar to the first component of GPCA and correctly follows a linear combination of $X_A$, but the $p$-values in Table 1 show that the causal component is not isolated in the first component. Finally, TLPLS can recover components which are Granger causal with $Y$, but without focusing on $X_A$ alone; instead, the first component is obtained as a linear combination of $X_A$ together with $X_C$ and $X_D$, which are *downstream* from $X_A$ in the causal graph and thus are obviously correlated with $Y$; therefore, TLPLS here is not able to discern causality from correlation.

**On high-dimensionality and nonlinear problems** We repeat the same experiments with a high-dimensional version of the same system, by setting $d = 500$ and thus $X_A \in \mathbb{R}^{100}$. We use again 1000 samples that lead to similar results, but

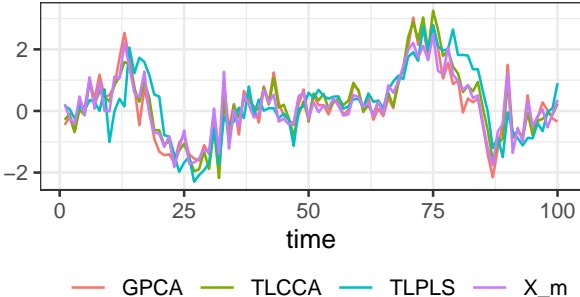

Figure 3: Extracted components for the considered methods ($d = 10$) together with the average of the first two coordinates of $X(t)$. All processes are centered and scaled.

exacerbated the issues observed previously for the competing methods: lack of causal meaning or causal ordering in the decomposition (see Figs. 5 and 6 in Appendix B.2). In particular, the proposed Granger-rotated PCA is the only method to clearly isolate the $X_A$ components. In the Appendix B.2.1 we also report results with a non-linear kernel PCA version in the same simulated example. We show that linear methods are not capable of fully extracting meaningful representations.

## 4.2 EFFECTS OF ENSO ON VEGETATION

Studying long-range spatio-temporal relations ('teleconnections') in the Earth system is paramount for understanding and modeling processes and interactions, but also to anticipate and forecast extreme events, finally attributing their causes. We consider here a classical example from Earth system science: the study of the long-range teleconnection between the El Niño Southern Oscillation (ENSO) -in particular the ENSO3.4 index- and vegetation greenness represented with the Normalized Difference Vegetation Index (NDVI) over Africa.

The NDVI was computed from MODIS reflectance data derived from the MCD43A4.006 BRDF- Adjusted Reflectance 16-Day L3 Global 500m product Schaaf and Wang [2015], Schaaf et al. [2002], which are disseminated from the Land Processes Distributed Active Archive Center (LP DAAC) also available at Google Earth Engine (GEE). We computed the NDVI at $8\,\mathrm{d}$ temporal and $0.5°$ spatial scales over 2007-2017 (11 years) and missing values were filled with linear interpolation. ENSO34 climate index was obtained from the Royal Netherlands Meteorological Institute (KNMI), the index is calculated daily based on Sea Surface Temperature (SST) anomalies averaged across the central equatorial Pacific Ocean (5N-5S, 170W-120W). ENSO3.4 time series were resampled to match NDVI temporal resolution by a sliding-window average filter.

We apply standard PCA, kernel PCA, and the correspond-

ing Granger-rotated versions to the NDVI observations ($n = 506$ time points for each of the $d = 11719$ non-water pixels). We consider a maximum lag of 10 (equivalent to 80 days) and retain only 10 components for each method. All methods consider normalized data with centered variables and unit variances. From the complete $p$-values of the Granger causality test for each component (see Table 4) we observe that standard principal component analysis obtain various components which are probably causally related with ENSO: PC3 ($p$-value $8.04 \times 10^{-4}$), PC6 and PC7 ($p$-values less then $0.1$). The Granger rotated PCA manage to concentrate the causally related components into the first two: GPCA1 ($p$-value $3.73 \times 10^{-3}$) and GPCA2 ($p$-value $5.81 \times 10^{-3}$), while the rest of the components are not found to be causally related with the ENSO index.

Table 2: Granger causality test $p$-values for each of the first five components ($d = 500$).

|    | PCA | GPCA | TLCCA | TLPLS |
|----|-----|------|-------|-------|
| C1 | 5.46e-78 | 0.00e+00 | 0.00e+00 | 1.49e-156 |
| C2 | 6.21e-04 | 8.64e-03 | 1.08e-207 | 7.52e-154 |
| C3 | 5.39e-146 | 3.11e-01 | 2.35e-32 | 9.52e-60 |
| C4 | 3.15e-02 | 3.59e-01 | 3.26e-25 | 1.16e-130 |
| C5 | 1.74e-128 | 3.06e-01 | 6.48e-31 | 2.05e-08 |

To visualize the spatial pattern associated with each component, we map in Fig. 4 the coefficients of the linear model for regressing the NDVI observations at each spatial location against the components estimated with a given method. Such maps depict the relationship between each component and NDVI at different locations. We can observe known patterns of vegetation response to NDVI, especially in East Africa and Sudan. Recent works have argued that El Niño led to drier than the average conditions in these regions, which impacted crop production in subsequent years. GPCA captures such patterns in a more meaningful and spatially coherent way [Kogan and Guo, 2017, Philippon et al., 2014, Kalisa et al., 2019].

## 5 CONCLUSIONS

We proposed a simple procedure to obtain a rotation that maximizes the Granger causality statistic between a fixed known cause and a projected principal mode of a high-dimensional temporal process. Such rotation can, in general, be applied as a post-processing step after ordinary PCA is used to reduce the observations to a lower-dimensional space. The Granger-rotated PCA concentrates in the first components the ones maximally related to the given cause signal. We derived an efficient procedure to compute the full Granger rotation and demonstrated its wide applicability with both simulated data and a real-world example involving high-dimensional spatio-temporal data.

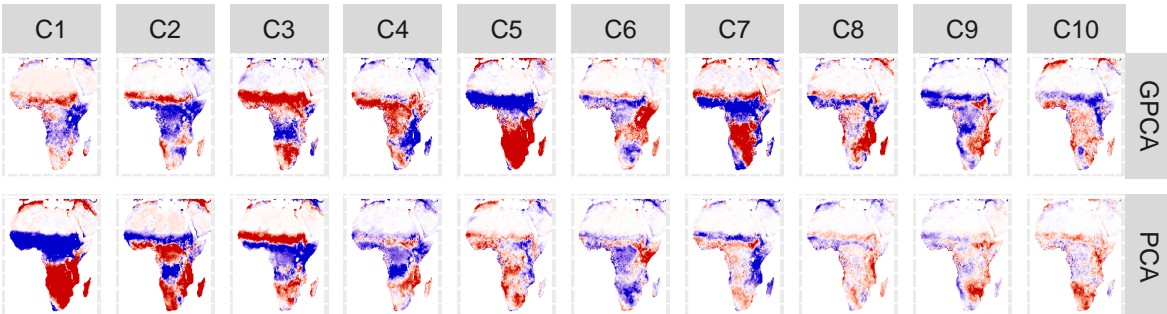

Figure 4: Relationship between the extracted components and the original NDVI data. The strength and sign are estimated by the corresponding coefficient in the linear regression of the NDVI values against the estimated components. Coefficients are normalized in each map and each component is sign-corrected to be positively correlated with the ENSO3.4 index.

The proposed GPCA is both simple and effective in recovering causally aligned principal components from high-dimensional data. Moreover, its similarity with established rotated PCA methods would suggest its possible applicability in a wide range of applications. Besides, it has not escaped our notice that alternative dimensionality reduction methods could benefit from the introduced methodology.

## Acknowledgements

This work was supported by the European Research Council (ERC) Synergy Grant "Understanding and Modelling the Earth System with Machine Learning (USMILE)" under Grant Agreement No 855187.

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

# A  GRANGER COMPONENTS

We denote with $X_{(j)}$ the $j$-th column of matrix $X$. To describe and fit auto-regressive (AR) models we define the following lagged sub-matrices of observations:

$$\mathbb{R}^{(n-m)\times d} \ni X^\ell = \begin{bmatrix} X_{m-\ell+1} \\ \vdots \\ X_{n-\ell} \end{bmatrix} \quad \text{for } \ell = 0, 1, \ldots, m.$$

And similarly for $Y$. Moreover we define

$$X_{\text{past}} = \begin{bmatrix} X^1 & \ldots & X^m \end{bmatrix}, \; Y_{\text{past}} = \begin{bmatrix} Y^1 & \ldots & Y^m \end{bmatrix},$$

and

$$[XY]_{\text{past}} = [X_{\text{past}} \; Y_{\text{past}}].$$

For each of the above definitions we define the corresponding ones for the transformed variables $\tilde{X}$, e.g. $\tilde{X}^\ell = X^\ell U$ and $\tilde{X}_{\text{past}} = \begin{bmatrix} X^1 U & \ldots & X^m U \end{bmatrix}$.

**First component.**  For the first component $\tilde{X}_{(1)}$, the RSSs of the two models are defined as follows:

$$RSS_0 = \left\| \tilde{X}^0_{(1)} - \tilde{X}_{\text{past}} \left( \tilde{X}^T_{\text{past}} \tilde{X}_{\text{past}} \right)^{-1} \tilde{X}^T_{\text{past}} \tilde{X}^0_{(1)} \right\|^2_2$$

$$RSS_1 = \left\| \tilde{X}^0_{(1)} - [\tilde{X}Y]_{\text{past}} \left( [\tilde{X}Y]^T_{\text{past}} [\tilde{X}Y]_{\text{past}} \right)^{-1} [\tilde{X}Y]^T_{\text{past}} \tilde{X}^0_{(1)} \right\|^2_2$$

Since $U$ is an orthogonal transformation we have that the residuals can be computed using the untransformed data matrix $X_{\text{past}}$ instead, and thus we can write them as a function of the first column of $U$ as,

$$RSS_0 = \left\| \left( I - X_{\text{past}} \left( X^T_{\text{past}} X_{\text{past}} \right)^{-1} X^T_{\text{past}} \right) X^0 U_{(1)} \right\|^2_2$$
$$= U^T_{(1)} (X^0)^T W^T W X^0 U_{(1)}$$
$$RSS_1 = U^T_{(1)} (X^0)^T V^T V X^0 U_{(1)}$$

where $W = I - X_{\text{past}} \left( X^T_{\text{past}} X_{\text{past}} \right)^{-1} X^T_{\text{past}}$ and $V = I - [XY]_{\text{past}} \left( [XY]^T_{\text{past}} [XY]_{\text{past}} \right)^{-1} [XY]^T_{\text{past}}$. We thus define the first column of $U$ as the solution of the following optimization problem for $RSS_0 - RSS_1$,

$$U_{(1)} = \arg \max_{||u||=1} u^T (X^0)^T \left( W^T W - V^T V \right) X^0 u.$$

Similarly to classical PCA, we ca recognize a Rayleigh quotient if we divide by $u^T u = 1$. And thus the maximum value is the leading eigenvalue of $(X^0)^T \left( W^T W - V^T V \right) X^0$ attained at the corresponding eigenvector.

The first Granger causal component is thus defined as $X(t) U_{(1)}$ and by definition it is the linear combination of the multi-dimensional process $X(t)$ for which $Y$ is *most Granger causal*.

**Additional components.**  As in PCA we can subtract the first $k$ components and repeat the process above to obtain the $k+1$ column of $U$. The resulting $U$ is composed of columns which are the eigenvectors of $(X^0)^T \left( W^T W - V^T V \right) X^0$ ordered by the corresponding eigenvalues.

**Simplification**  We show here that the relatively complex expression for $W$ and $V$ can be simplified and the Granger components can be obtained similarly to PCA with an additional cost of a QR decomposition of the matrix $[XY]_{\text{past}}$.

Let $QR = [XY]_{\text{past}}$ be the QR decomposition [1] of $[XY]_{\text{past}}$, with $Q$ orthogonal and $R$ upper triangular. Assume $n - m > md + mp$, then we have that $Q = [Q_1 Q_2 Q_3]$; where $Q_1 \in \mathbb{R}^{(n-m)\times md}$ and $Q_2 \in \mathbb{R}^{(n-m)\times mp}$. Thus,

$$[XY]_{\text{past}} = [Q_1 Q_2 Q_3] \begin{bmatrix} R_1 & R_{21} \\ 0 & R_2 \\ 0 & 0 \end{bmatrix}$$
$$= [Q_1 R_1 \; Q_2 R_2 + Q_1 R_{21}].$$

That is, $Q_1 R_1$ is the compact QR decomposition for $X_{\text{past}}$. It is easy to see now that $W = Q_2 Q_2^\top + Q_3 Q_3^\top$ and $V = Q_3 Q_3^\top$. Thus, by the orthogonality of $Q = [Q1 Q2 Q3]$ we have that,

$$W^\top W - V^\top V = Q_2 Q_2^\top$$

*Proof.* First we prove that $W = Q_2 Q_2^\top + Q_3 Q_3^\top$,

$$W = I - X_{\text{past}} \left( X^T_{\text{past}} X_{\text{past}} \right)^{-1} X^T_{\text{past}}$$
$$= QQ^\top - Q_1 R_1 \left( R_1^\top Q_1^\top Q_1 R_1 \right)^{-1} R_1^\top Q_1^\top$$
$$= Q_1 Q_1^\top + Q_2 Q_2^\top + Q_3 Q_3^\top - Q_1 R_1 R_1^{-1} R_1^{-t} R_1^\top Q_1^\top$$
$$= Q_2 Q_2^\top + Q_3 Q_3^\top$$

Similarly $V = Q_3 Q_3^\top$. Now we just need to observe that

$$W^\top W = Q_2 Q_2^\top Q_2 Q_2^\top + Q_2 Q_2^\top Q_3 Q_3^\top$$
$$+ Q_3 Q_3^\top Q_2 Q_2^\top + Q_3 Q_3^\top Q_3 Q_3^\top$$
$$= Q_2 Q_2^\top + Q_3 Q_3^\top$$

since $Q_i^\top Q_i = I$ and $Q_i^\top Q_j = 0$ for $i \neq j$. Similarly $V^\top V = Q_3 Q_3^\top$. $\square$

**Solving with SVD**  For numerical stability it is preferable not to compute the spectral decomposition directly, but instead rely on the singular value decomposition (SVD). To do so, we can simply obtain the SVD of the transformed data matrix $Q_2^\top X^0$.

---

[1] without using pivoting

# B EXPERIMENTS DETAILS

## B.1 IMPLEMENTATION AND PACKAGES

We used the PCA and CCA implementations available in the `stats` package distributed with R [R Core Team, 2022]. For PLS we used the implementation available in the `ropls` package [Thevenot et al., 2015]. The kernel PCA was computed via the implementation in the `kernlab` package [Karatzoglou et al., 2004]. Additional packages used were: `ggplot2` [Wickham, 2016] and `ggspatial` [Dunnington, 2021] for plotting; `raster` [Hijmans, 2022] and `ncdf4` [Pierce, 2021] for data handling.

## B.2 SIMULATED DATA

We detail here the complete structural equations of the dynamical system used to simulate the data in Section 4.1. We consider the following data generating process where $X \in \mathbb{R}^d$ and $|A| = \ldots = |E| = d/5$,

$$Z_1(t) = \cos\left(\frac{4}{t}10\right) + \eta_1,$$

$$Z_2(t) = \cos\left(\frac{5t}{100}t\right) + \eta_2,$$

$$Y(t) = \sin\left(\frac{2}{10}t\right) + \cos\left(\frac{t}{10}\right) + \varepsilon,$$

$$X_A(t) = \frac{2}{10}X_A(t-2) + \frac{4Y(t-1) + 3Y(t-2) + 2Y(t-3)}{10}$$
$$+ \frac{2}{10}\left(Z_1(t-1) + Z_1(t-2)\right) + \varepsilon_A,$$

$$X_B(t) = \frac{4}{10}X_B(t-1) + \frac{5}{10}Z_1(t-1) + \varepsilon_B,$$

$$X_C(t) = \frac{1}{10}X_C(t-2) + \frac{8}{10}Z_1(t-2)$$
$$+ \frac{8}{10}\overline{X_A}(t-1) + \frac{4}{10}\overline{X_A}(t-2) + \varepsilon_C,$$

$$X_D(t) = \frac{1}{10}X_D(t-2) - \frac{7}{10}Z_2(t-3) + \frac{8}{10}\overline{X_C}(t-2) + \varepsilon_D,$$

$$X_E(t) = \frac{1}{10}X_E(t-1) + \frac{8}{10}Z_1(t-2)$$
$$+ \frac{6}{10}Z_2(t-1) + \frac{2}{10}Z_2(t-2) + \varepsilon_E,$$

where $\eta_1, \eta_2, \varepsilon, \varepsilon_i$ are all independent standard Gaussian noise.

### B.2.1 Non-Linear Simulated Data

We also consider a non-linear version of the same dynamical structural equation models depicted in Figure 1. We simply replace the linear functions for trigonometric polynomials of degree two.

We apply the same methods plus the kernelized version kG-PCA. Results of the Granger causality test for each learned

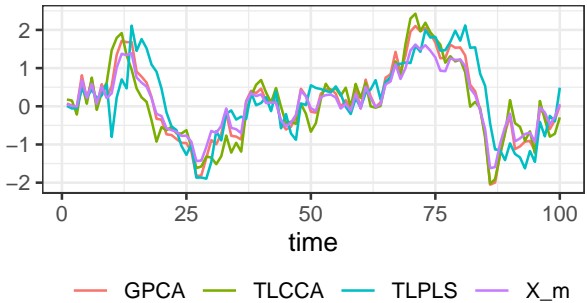

Figure 5: Causally related components for the considered methods together with the average of the first two coordinates of $X(t)$ ($d = 500$). All processes are centered and scaled.

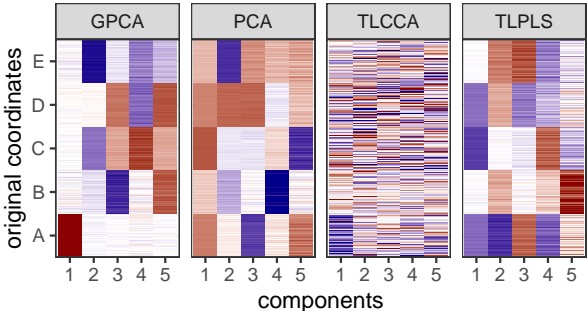

Figure 6: Rotations learned with the various methods over the simulated data ($d = 500$).

components are reported in Table 3 and the rotation matrix for the linear case is depicted in Figure 7. We can observe that the non-linear version is able to recover a more Granger related component and from the rotation matrices in Figure 7 we see that none of the linear methods is able to correctly isolate $X_A$ components in one of the learned signals.

Table 3: Granger causality test $p$-values for each component of the different methods. Non-linear simulated data in the high-dimensional setting ($d = 500$).

|    | PCA | GPCA | kGPCA | TLPLS |
|----|---------|---------|---------|---------|
| C1 | 2.09e-02 | 4.58e-02 | 1.25e-02 | 3.81e-02 |
| C2 | 5.55e-01 | 8.44e-01 | 4.53e-01 | 3.77e-07 |
| C3 | 5.70e-01 | 2.84e-01 | 1.02e-01 | 2.14e-21 |
| C4 | 3.50e-01 | 9.77e-01 | 9.51e-01 | 3.00e-02 |
| C5 | 5.51e-01 | 9.92e-01 | 9.84e-01 | 3.10e-05 |

## B.3 ENSO - NDVI IN AFRICA

We report here additional results for the application of Granger-rotated (k)PCA to the study of the effects of ENSO

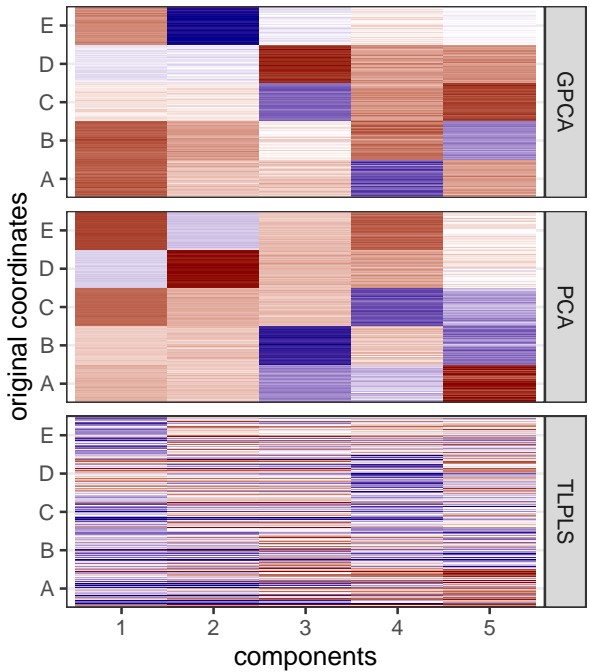

Figure 7: Rotations learned with the various methods over the non-linear simulated data ($d = 500$).

on vegetation in Africa. In particular, Table 4 shows the $p$-values of the Granger causality test using the learned components by both linear and nonlinear methods. As in the simulation experiments, we also observe here that GPCA, and kGPCA, concentrate the most Granger-causally relevant components. This addresses the challenge of component selection.

Table 4: Granger causality test $p$-values for each component of the different methods, both linear and nonlinear.

|  | PCA | GPCA | kPCA | kGPCA |
|---|---|---|---|---|
| C1 | 8.71e-02 | 3.73e-03 | 2.92e-01 | 2.45e-04 |
| C2 | 6.10e-02 | 5.81e-03 | 6.31e-01 | 3.61e-02 |
| C3 | 8.04e-04 | 3.79e-01 | 7.59e-01 | 1.43e-01 |
| C4 | 1.18e-01 | 4.36e-01 | 6.02e-01 | 5.42e-01 |
| C5 | 1.82e-01 | 1.86e-01 | 9.51e-01 | 8.04e-01 |
| C6 | 5.09e-02 | 9.28e-01 | 3.01e-01 | 8.61e-01 |
| C7 | 2.62e-02 | 9.59e-01 | 5.85e-03 | 9.95e-01 |
| C8 | 7.11e-01 | 1.00e+00 | 4.87e-01 | 9.99e-01 |
| C9 | 3.42e-01 | 1.00e+00 | 1.39e-01 | 1.00e+00 |
| C10 | 6.46e-01 | 1.00e+00 | 1.09e-01 | 1.00e+00 |

The learned components of GPCA along competing methods are shown in Fig. 8. The top GPCA component is more correlated with the ENSO signal than the third one from PCA (actually the first two were related to trivial, non-informative seasonal and annual cycles).

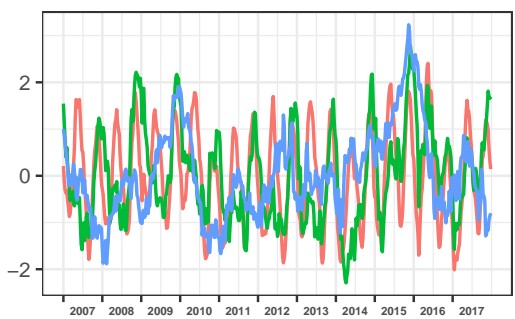

Figure 8: Learned components PCA3 (red) and GPCA1 (green) which are found to be causally related with ENSO (in blue).