# OpenReview forum: "Learning Causal Representations with Granger PCA"
_auai.org/UAI/2022/Workshop/CRL — CRL@UAI 2022 Poster_

### Official Review · Reviewer_qVJQ · 2022-06-19
**Identifying high-level effect variables of known, low-dimensional causes**

**Rating:** 6
**Confidence:** 4

**Review:**

This paper describes an extension of PCA to identify features from a high-dimensional input that are effects of a known cause variable. The proposed method, Granger PCA, does this by finding an orthogonal matrix for which the first dimensions maximize the predictive power of the cause.

The method is in general well presented, although some parts has to be read twice to fully understand. For instance, in the beginning of Section 2.2, it would be beneficial to give some intuition behind the approach and more description on the Granger-cause equations. Furthermore, the main task of the paper it missing in the abstract: "learning high-level representations that are effects of known low-dimensional causes." The abstract currently only discusses "learning Granger-causal feature representations" which is too high-level.

With respect to learning effects, it is unclear from the paper whether this only considers direct effects, as X_A in the simulation experiment section 4.1, or also indirect effects, for instance if Y → X_A → X_C without the latent confounder Z_1.

In terms of experiments, the presentation of the results is unclear in some parts. For instance, in Figure 2, there is no legend explaining what the different colors mean. Similarly, the p-values in Table 1 are difficult to interpret. What would be the optimal scores we aim for, especially for the components besides C1? How do the indirect effects of Y to X_C and X_D influence the results?

In summary, the paper proposes a relevant method for finding effects in high-dimensional data of known low-dimensional causes. While the presentation of the paper needs to be improved, I believe that this is possible for the camera-ready version.

Minor comments:
- Page 2, Setion 2.3: there seems to be a typo in the dimensions of $Y$ ($Y \in \mathbb{R}^{n\times p}$ instead of $Y \in \mathbb{R}^{n\times d}$)
- Page 1, Section 1: the second paragraph about causal representation learning is very sparse in citations. For instance, to point out the difference task to this paper, discussing the goal of other CRL methods puts this paper into a better context.

---

### Official Review · Reviewer_9Xc5 · 2022-06-24
**Easy-to-use PCA variation to detect Granger-causal variables, but theoretical analysis could be stronger**

**Rating:** 6
**Confidence:** 3

**Review:**

**Quality**: The method proposed is a sensible modification of PCA to take into account Granger causality. The approach is quite simple and intuitive. However, the paper does not really justify it theoretically, and I am left wondering under which circumstances it does work. Is there an identifiability guarantee? What assumptions are needed? Can the method deal with confounding?

The authors do validate that the method works in experiments, both on toy data and a real-life example, but more diverse experiments could be useful to find out when it works and when it doesn't.

**Clarity**: The paper is clearly structured and generally well-written. It mostly focuses on the somewhat narrow problem of linear Granger causality. In my opinion, the paper would benefit from an extended comparison to non-linear methods beyond kernel PCA, and to other flavours of causality and causal representation learning. Some strong statements like "PCA is undoubtedly the preferred method in real practice" may be true in some fields, but are dubious in the stated generality.

**Originality**: I am not an expert on Granger causality and cannot really judge the novelty of the proposed method.

**Significance**: The method could be applicable to plenty of real-world time series, as the authors demonstrate convincingly with the ENSO dataset. Its simplicity might enable many researchers to apply it to their problems. However, since the paper does not fully characterize under which conditions the method is expected to work, I find it difficult to really estimate how often its outputs will be useful.

**Pros**:
- The proposed method seems remarkably easy to use.
- The paper is clearly structured and generally well-written.
- The empirical demonstration on both toy data and a real-world example from the earth sciences is nice.

**Cons**:
- When and why does this method work? The paper does not really provide any theoretical guarantees (like an identifiability theorem), and the empirical evaluation is not detailed enough to probe the limits of the applicability of the method.
- The paper is quite focussed on the linear case, the non-linear setting deserves more discussion besides a pointer to kernel PCA.
- It would be beneficial if the authors could position their approach relative to the many recently proposed methods for (often neural) causal representation learning.

**Summary**: With Granger PCA, the authors introduce a simple and intuitive method and convincingly demonstrate it in experiments. While I wish for a more thorough (theoretical or empirical) analysis of why and when it works, I believe that this paper makes for a good workshop paper.

---

### Meta-Review · Program_Chairs · 2022-07-06

**Recommendation:** Accept (Poster)
**Confidence:** 3

**Metareview:**

While the reviewers have the impression that more theoretical and empirical characterisation of the method may strengthen the contribution, they also both agree that the work may be of interest for the workshop. The authors are encouraged to incorporate the suggestions of the two reviewers for a future version.

---

### Decision · Program_Chairs · 2022-07-06

Accept (Poster)